# POMDIFFUSER: LONG-MEMORY MEETS LONG-PLANNING FOR POMDPS

## ABSTRACT

Effective long-term planning in complex environments benefits from not only leveraging immediate information but also utilizing past experiences. Drawing inspiration from how humans use long-term memory in decision-making, we propose the POMDiffuser framework, an approach to planning in partially observable environments. While conventional Diffuser models often memorize specific environments, POMDiffuser explores the potential of learning to plan from memory, with the aim of generalizing to new scenarios. By incorporating a memory mechanism in POMDP scenarios, our model extends diffusion-based planning models into the realm of meta-learning with carefully designed tasks that require the diffusion planner to demonstrate both long-term planning and memory utilization. We investigated existing diffusion-based models, focusing on their applicability, computational efficiency, and performance trade-offs.

## 1 INTRODUCTION

To operate effectively in complex environments, an intelligent agent must have two key abilities: the capacity to memorize past experiences and the ability to use this memory to imagine future scenarios for planning (Schacter et al., 2007; 2012). These abilities are particularly crucial in partially observable environments (Kaelbling et al., 1998) where current observations lack sufficient information for optimal decision-making. In such scenarios, agents need to infer the hidden state of the world—known as the belief state—by leveraging past experiences.

For both memory and planning, the critical challenge lies in extending the horizon length both backward and forward. The quality of the belief state relies heavily on how far into the past the agent can consider, as a longer history provides a richer belief representation. Similarly, the advantages of long-horizon planning—such as avoiding short-sighted decisions, aligning immediate actions with long-term objectives, addressing sparse reward issues, and effectively managing unfamiliar tasks—become more pronounced as the planning range extends (Hamrick et al., 2020). In particular, achieving long-term memory and extended planning simultaneously is critical (Momennejad, 2024; Gregor et al., 2019).

Although various model architectures have been explored to enhance memory and planning abilities, these architectures face significant challenges in effectively integrating long-horizon memory with long-horizon planning. Recurrent Neural Networks (RNNs) significantly limit their scalability with large datasets as its training process does not allow parallel processing of a sequence. Their dependence on autoregressive rollouts for planning leads to error compounding, which worsens in particular with longer planning horizons (Lambert et al., 2022). Lastly, the vanishing gradient restricts their memory to retain long-term dependencies.

Transformers (Vaswani et al., 2017) have emerged as an alternative, capturing dependencies over long sequences without sequential processing constraints. It also excels in parallel computation and capturing global context. However, each step of the rollouts involves quadratic complexity with respect to sequence length, making them computationally intensive for extended planning tasks. Furthermore, they still face challenges due to their reliance on autoregressive rollouts, leading to compounding error (Lambert et al., 2022; Bachmann & Nagarajan, 2024).

Structured State Space Models (SSMs) , such as Mamba (Gu & Dao, 2024), offer a promising alternative to the intensive computation of Transformers by modeling long sequences with linear

complexity relative to sequence length while preserving parallel trainability. Although SSMs reduce the cost of a single-step rollout to constant complexity $O(1)$ compared to the quadratic complexity of Transformers, they still rely on autoregressive planning.

The Diffuser (Janner et al., 2022; Ajay et al., 2023) approach, a new planning method based on Diffusion Models (Sohl-Dickstein et al., 2015; Ho et al., 2020), has recently emerged as a promising paradigm in planning. It addresses the compounding error issue by generating the entire sequence simultaneously, treating the sequence like an image, which allows for holistic sequence generation. This approach enables more accurate and efficient planning over long horizons. However, as noted by the authors of Decision Diffuser (Ajay et al., 2023), a major limitation of diffusion-based planning is that it has so far been applicable only to MDP settings. Extending this approach to POMDPs—where long-context memory must be effectively integrated—remains an open challenge.

In this paper, we address these limitations by conducting the first systematic investigation of long-memory, long-planning diffusion models for POMDPs. To this end, we introduce a diffusion planning framework called *POMDiffuser*, which integrates different versions of POMDiffusers based on the belief encoding architecture used alongside the Diffuser planner. Specifically, we explore POMDiffusers built on RNNs, Transformers, and SSMs, analyzing the strengths and weaknesses of each approach, particularly in terms of achieving both long-memory and long-planning capabilities. Furthermore, by encoding and conditioning on the belief representation, this framework offers a natural extension of the Diffuser planner as a meta-planner.

Additionally, as no benchmark currently exists to evaluate long-memory and long-planning capabilities within the Diffusion framework, we propose a new benchmark suite to fill this gap. Our experimental results demonstrate that SSM-POMDiffuser performs well in tasks requiring complex reasoning ability from the long and global contextual memory, in planning problems, while enjoying superior computational efficiency. However, we found that it struggled with more complex long-memory and long-horizon planning tasks, where the agent must remember detailed aspects of the environment.

The contributions of this paper are: **(i)** the first systematic empirical investigation of long-memory, long-planning diffusion models for POMDPs, **(ii)** the introduction of the POMDiffuser framework, which for the first time extends the Diffuser planner's capabilities to POMDPs, and **(iii)** the development of a benchmark suite for Diffusion Planning in POMDPs.

## 2 BACKGROUND

### 2.1 STATE SPACE MODELS

Structured State Space Models (SSMs) are sequence-to-sequence models well-suited for tasks that require significant memory retention and are particularly effective at processing long sequences due to their computational efficiency. These models transform an input sequence $\mathbf{x}_{1:T} \in \mathbb{R}^{T \times D}$ into an output sequence $\boldsymbol{y}_{1:T} \in \mathbb{R}^{T \times D}$ through a specific recurrence relation:

$$
\begin{aligned}
\boldsymbol{h}_t &= \boldsymbol{A}_t \boldsymbol{h}_{t-1} + \boldsymbol{B}_t \mathbf{x}_t, \\
\boldsymbol{y}_t &= \boldsymbol{C}_t \boldsymbol{h}_t.
\end{aligned}
\tag{1}
$$

At each time step $t$, $\boldsymbol{x}_t$ and $\boldsymbol{y}_t$ both belong to $\mathbb{R}^D$, representing the input and output at that moment. The hidden state $\boldsymbol{h}_t \in \mathbb{R}^H$ captures the historical information up to time $t$. The matrices $\boldsymbol{A}_t \in \mathbb{R}^{H \times H}$, $\boldsymbol{B}_t \in \mathbb{R}^{H \times D}$, and $\boldsymbol{C}_t \in \mathbb{R}^{D \times H}$ are designed to model long-range dependencies within the sequence efficiently. In time-invariant SSMs, where $\boldsymbol{A}_t$, $\boldsymbol{B}_t$, and $\boldsymbol{C}_t$ remain constant, $\boldsymbol{y}_{1:T}$ can be computed in parallel from $\mathbf{x}_{1:T}$, enhancing training efficiency. $\boldsymbol{A}_t$ is often diagonal or block-diagonal, with its eigenvalues initialized near the unit circle to facilitate stability over long sequences (Gu et al., 2020; 2022). Recent studies have explored conditioning these matrices on the input sequence, allowing the model to adapt and focus on pertinent input information (Gu & Dao, 2024).

### 2.2 DIFFUSION PROBABILISTIC MODELS FOR PLANNING

Diffusion probabilistic models (Sohl-Dickstein et al., 2015; Ho et al., 2020; Song & Ermon, 2021) have achieved remarkable success in various image generation tasks (Dhariwal & Nichol, 2021;

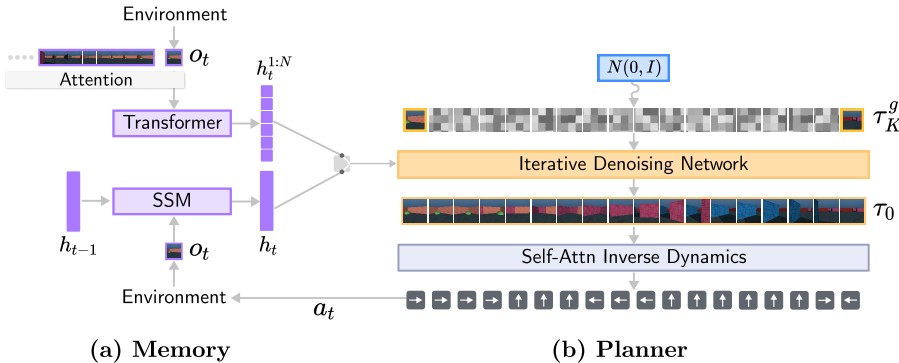

(a) Memory          (b) Planner

Figure 1: **Overview of POMDiffuser in the inference stage.** When using the Transformer-based memory, it achieves more accurate memory-aligned planning but suffers from quadratic computation, whereas SSM memory benefits from constant time complexity by updating the current belief.

Rombach et al., 2022; Ramesh et al., 2022; Saharia et al., 2022; Liu et al., 2024). These models generate data by iteratively denoising across $K$ steps, starting from Gaussian noise $\mathbf{x}_M \sim \mathcal{N}(\mathbf{0}, \mathbf{I})$. The generative process is expressed as:

$$p_\theta(\mathbf{x}_0) = \int p(\mathbf{x}_K) \prod_{k=0}^{K-1} p_\theta(\mathbf{x}_k \mid \mathbf{x}_{k+1}) \, \mathrm{d}\mathbf{x}_{1:K}, \tag{2}$$

where each transition $p_\theta(\mathbf{x}_k \mid \mathbf{x}_{k+1})$ is a Gaussian with learnable mean $\boldsymbol{\mu}_\theta(\mathbf{x}_{k+1})$ and fixed covariance $\sigma_k^2 \mathbf{I}$. The model is trained to predict the noise $\boldsymbol{\epsilon}$ added to the data $\mathbf{x}_0$ during the forward diffusion process:

$$\mathcal{L}(\theta) = \mathbb{E}_{\mathbf{x}_0, k, \boldsymbol{\epsilon}} \left[ \| \boldsymbol{\epsilon} - \boldsymbol{\epsilon}_\theta(\mathbf{x}_k) \|^2 \right], \tag{3}$$

where $\mathbf{x}_k = \sqrt{\bar{\alpha}_k} \mathbf{x}_0 + \sqrt{1 - \bar{\alpha}_k} \boldsymbol{\epsilon}$ and $\boldsymbol{\epsilon} \sim \mathcal{N}(\mathbf{0}, \mathbf{I})$. Building on this framework, Janner et al. (2022) introduces *Diffuser*, a diffusion-based model for planning in offline reinforcement learning settings. Trajectories of states and actions are formatted into a two-dimensional array:

$$\tau = \begin{bmatrix} \mathbf{s}_0 & \mathbf{s}_1 & \dots & \mathbf{s}_T \\ \mathbf{a}_0 & \mathbf{a}_1 & \dots & \mathbf{a}_T \end{bmatrix}. \tag{4}$$

Diffuser uses a diffusion model $p_\theta(\tau)$ to generate complete trajectories. It efficiently plans long sequences, avoiding the cumulative errors common in other planning approaches.

## 3   MEMORIZE LONG TO PLAN LONG

### 3.1   MEMORY

To build an efficient model for long-memory and long-horizon planning tasks, POMDiffuser consists of two main components: memory and planner. Due to its flexible conditioning methodology, it can integrate various memory architectures, but two main candidates stand out. The first is recurrent memory, which offers constant time complexity during inference *(RNN-POMPDiffuser)*. However, RNN-based memory make a training time bottleneck that isn't from the Diffusion based planner. Thus, SSM-based memory is more practical, as it provides both constant time complexity for belief updates and parallelizable training. We instantiated this as *SSM-POMDiffuser*, using an SSM as the memory encoder:

$$\mathbf{h}_t \leftarrow f_{\text{recurrent}}(\mathbf{h}_{t-1}, \mathbf{o}_t, \mathbf{a}_{t-1}) \tag{5}$$

where $\mathbf{o}_t$ is the current observation from the environment and $\mathbf{a}_{t-1}$ is the previous action. In addition, $f_{\text{recurrent}}$ can be any memory model that recursively models $p(\mathbf{x}_{1:T})$, e.g. GRU Chung et al. (2014) or Mamba Gu & Dao (2023). While it is common to encode additional information such as the reward $\mathbf{r}_t$ and done signal $\mathbf{d}_t$ when relevant to the task, we omit this information since we are considering a setup with sparse rewards, where modeling world dynamics is more crucial.

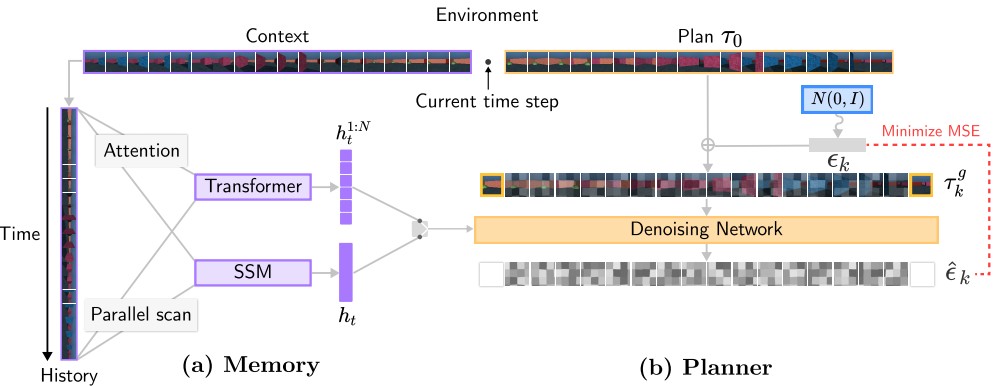

Figure 2: **Overview of POMDiffuser in the training stage.** We trained our model in an offline RL setting, where we sampled batches of context and plan pairs.

Another attractive option is Transformer-based memory (Parisotto et al. (2020)), which excels due to its powerful pairwise interactions between tokens using attention mechanism. In *Transformer-POMDiffuser*, memory is no longer a compressed embedding but a set of tokens.

$$\mathbf{h}_t^{1:N \times 2} \leftarrow f_{\text{Tranformer}}(\mathbf{o}_t^{1:N}, \mathbf{a}_t^{1:N}, \mathbf{p}) \tag{6}$$

where $\mathbf{p}$ is the positioning embedding vector. In a reinforcement learning setting, the input is a trajectory consisting of multiple sequences of observation and action pairs. Typically, the trajectory is truncated to the maximal length $N$ that the Transformer model can handle.

## 3.2 PLANNING WITH THE MEMORY

To condition contextual information when the Diffusion model generates data, there are two options: using external memory, as we propose, in a *heterogeneous* manner, or using a single Diffusion model by incorporating the past clean trajectory as part of the denoising target $\boldsymbol{\tau}_{1:t}^k \leftarrow \boldsymbol{\tau}_{1:t}^0$ during the denoising process, where $k$ is a random denoising step in the Diffusion modeling process. This *homogeneous* approach of modeling context and the generation process simultaneously may be simple and effective, but it inherently suffers from quadratic complexity.

We suggest detaching this process into two separate components—memory and planning—as it reduces the time complexity from $O((L+H)^2)$ to $O(L \log L + H^2)$, where $L$ is the memory length and $H$ is the planning horizon. Since we adopt the heterogeneous approach to modeling memory, it must be conditioned when the plan decoder generates a plausible trajectory. There are generally two ways to incorporate memory information in a heterogeneous manner: by concatenating the memory embedding with the noisy input, or through cross-attention computation during the denoising process. We chose the latter, as it allows for more computation during the denoising process and aligns well with the memory representation being a set of tokens in Transformer-POMDiffuser. After conditioning the memory in the denoising process, the memory and planner are jointly trained with the diffusion modeling objective.

$$\mathcal{L}(\theta) = \mathbb{E}_{\boldsymbol{\tau}^0, \boldsymbol{\epsilon}, \mathbf{h}_t} \left[ \left\| \boldsymbol{\epsilon} - \epsilon_\theta(\boldsymbol{\tau}^k, \mathbf{h}_t) \right\|^2 \right] \tag{7}$$

where $\boldsymbol{\tau}^k = \sqrt{\bar{\alpha}^k}\boldsymbol{\tau}_0 + \sqrt{1-\bar{\alpha}^k}\boldsymbol{\epsilon}, \quad \boldsymbol{\epsilon} \sim \mathcal{N}(\mathbf{0}, \mathbf{I})$. To avoid confusion in notation, we clarify that $\mathbf{k}$ refers to the denoising step variable, and $\beta$ represents the time step in the agent's environment. As the denoising decoder, we primarily adopted a Transformer decoder denoising network, while the UNet model was used only for the Superimposed-MNIST task. This is how the denoising network receives noisy input and predicts the noise in Transformer-decoder network:

$$\mathbf{z}^0 = f_{\text{MLP}}([\boldsymbol{\tau}^k, \mathbf{p}, \mathbf{k}]) \tag{8}$$

$$\mathbf{z}^{l+1} = f_{\text{Transformer}}^l(\mathbf{z}^l, \mathbf{h}) \tag{9}$$

where $\boldsymbol{\tau}^k$ is the input, $\mathbf{p}$ is the position embedding, $\mathbf{k}$ is denoising step embedding, $\mathbf{h}_1, \ldots, \mathbf{h}_{L-1}$ are the hidden states and $\hat{\epsilon}^k = \mathbf{h}_L$ is the output.

### 3.3 SELECTING ACTIONS THROUGH INVERSE DYNAMICS

When using diffusers for planning, relying solely on observations and employing inverse dynamics to deduce actions has proven to be effective (Ajay et al., 2023). This approach is particularly beneficial when states are continuous but actions are discrete (Tedrake). Based on this, we adopted the inverse dynamics model to predict actions from observations. However, unlike in MDPs, predicting actions solely from adjacent frames in POMDPs can be unreliable. To address this issue, we used Transformer encoders (Vaswani et al., 2017) to predict the full action sequence from the entire trajectory $\boldsymbol{\tau}_0$, expressed as:

$$\boldsymbol{\tau}_0 = (s_t, s_{t+1}, ..., s_{t+H-1}) , \qquad \mathbf{a}_{1:H} = f_{\text{Transformer}}(\boldsymbol{\tau}_0) . \qquad (10)$$

### 3.4 STRATEGIES TO PLAN LONGER

As the planning horizon increases, the computational burden on the Diffusion model grows quadratically. To address this issue, we introduce *latent-level planning*, inspired by work from other domains, where it has proven effective in generating and handling high-resolution datasets using VAEs (Rombach et al., 2022). We observe a similar challenge in long-horizon planning, where operating directly in the observation space becomes inefficient. To mitigate this, we demonstrate the efficacy of planning at the latent level using a pre-trained autoencoder, showing significant improvements in computational cost.

## 4 RELATED WORK

**Efficient World Models.** Model-based reinforcement learning (MBRL) is renowned for its sample efficiency, utilizing world models for planning or policy learning in imagination trajectories (Kalweit & Boedecker, 2017; Ha & Schmidhuber, 2018). Commonly, MBRL incorporates RNNs (Ha & Schmidhuber, 2018; Hafner et al., 2019a;b; 2020; 2023) or Transformers (Chen et al., 2021; Micheli et al., 2023; **?**) as its backbone architectures. However, despite its improved sample efficiency, MBRL's computational inefficiency is often limited by the constraints of the backbone architectures. To overcome these limitations, recent advancements have introduced State Space Model-based world models that enhance computational efficiency and sustain performance, especially in tasks requiring long-term memory (Deng et al., 2024; Momennejad, 2024). However, despite their advancements, these models still face challenges in long-horizon planning due to error accumulation in autoregressive modeling (Lambert et al., 2022; Bachmann & Nagarajan, 2024).

**Conditioned Diffusion for Planning.** Recent advances in conditional generative modeling have enabled diffusion models to generate high-quality outputs based on conditions (Ho et al., 2020; Saharia et al., 2022; Liu et al., 2024). In decision-making, these techniques guide diffusion-based planners using return values, tasks, or constraints to generate trajectories (Ajay et al., 2023; Ni et al., 2023; Liang et al., 2023; Chen et al., 2024). The Decision Diffuser Ajay et al. (2023) employs conditional generative modeling to replace traditional value function estimation with a return-conditioned diffusion model. Although effective in MDPs, it lacks demonstration for long-horizon planning in POMDPs, where maintaining long-range beliefs is crucial. Additionally, the method's quadratic increase in time complexity with contextual information makes it impractical for environments with extensive context requirements. MetaDiffuser (Ni et al., 2023) and AdaptDiffuser (Liang et al., 2023) showed how to plan in a heterogeneous manner but did not address long planning with long-term dependencies in an environment. Diffusion forcing (Chen et al., 2024) first demonstrated past history-conditioned plan generation using GRU memory in POMDPs, yet it did not conclusively address the feasibility of generating globally contextualized plans in environments requiring extensive memory. To our knowledge, our work is the first to focus on integrating long context for long-horizon planning.

## 5 EXPERIMENTS

### 5.1 ENVIRONMENTS

To evaluate POMDiffuser's performance in POMDPs with long-horizon planning and long memory, we designed three tasks: superimposed MNIST, 2D Memory Maze, and Blind Color Matching.

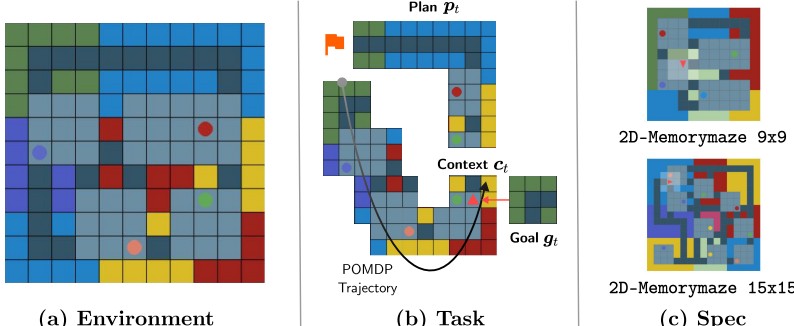

(a) Environment | (b) Task | (c) Spec

Figure 3: **2D Memory Maze**. (a) Procedurally-generated environment. (b) Goal-conditioned navigation task.

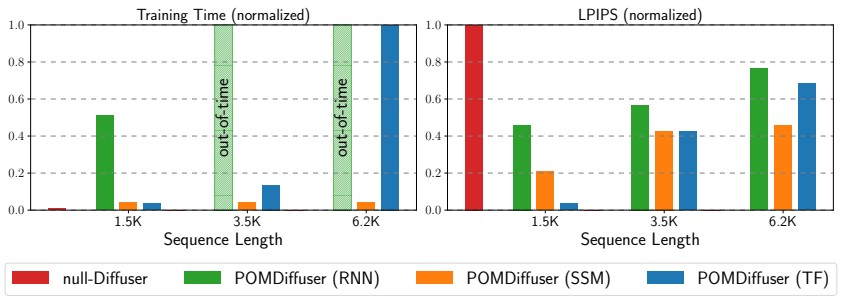

Figure 4: **Results of Superimposed MNIST.** While the Transformer model excels in short-memory tasks, the SSM model outperforms it in both training efficiency and performance as task length increases. Due to time constraints in processing 3.5K and 6.2K data with RNNs, we used specially designed datasets for RNN-POMDiffuser.

**Superimposed MNIST**   To demonstrate POMDiffuser's ability to memorize and plan in the simplest scenario, we designed a task using the MNIST dataset, where each image $x \in \mathbb{R}^{1 \times H \times W}$ is flattened into $\tilde{x} \in \mathbb{R}^{HW}$. Two flattened images $\tilde{x}_1$ and $\tilde{x}_2$ are randomly selected from the MNIST dataset. At each time step $t$, the model receives a single pixel value $o_t \in \mathbb{R}$, defined as follows:

$$o_t = \begin{cases} \tilde{x}_1^{(t)} & \text{if } 1 \le t \le L/2, \\ \tilde{x}_2^{(t-L)} & \text{if } L/2 < t \le L+1 \end{cases} \tag{11}$$

After observing all 1568 pixel values, the model predicts the pixel-wise superposition of the two images $y \in \mathbb{R}^{784}$. The target image is computed by: $y = \tilde{x}_1 + \tilde{x}_2$. This requires the model to memorize both images $\tilde{x}_1$ and $\tilde{x}_2$ and plan how to reconstruct their superposition.

**2D Memory Maze**   To evaluate the agent's performance in goal-conditioned planning tasks under minimal settings, we simplified the Memory Maze task from Pasukonis et al. (2022) while retaining its core features of strong partial observability and reward sparsity. In each episode, a procedurally generated map is created with randomized elements, including wall colors, goal locations, and grid layouts. The agent is restricted to observing only a partial top-down view of the map. The objective is to navigate from the current observation $\mathbf{o}_t$ to the target goal $\mathbf{o}^{\text{goal}}$.

Initially, the agent explores the environment to collect burn-in context frames by navigating the map. Subsequently, the agent is tasked with planning a path to the goal. Since the maze configuration changes every episode, the agent cannot memorize specific maps but must instead learn to infer the structure of the given map and the location of the goals based on episodic experience. For further details, refer to Appendix B.2.

**Blind Color Matching**   We extended our experiments to a robotics control task that requires long-term planning and memory. The task involves picking and placing distributed blocks onto floors that match each block's color. The robot agent receives a sparse reward only after placing all blocks

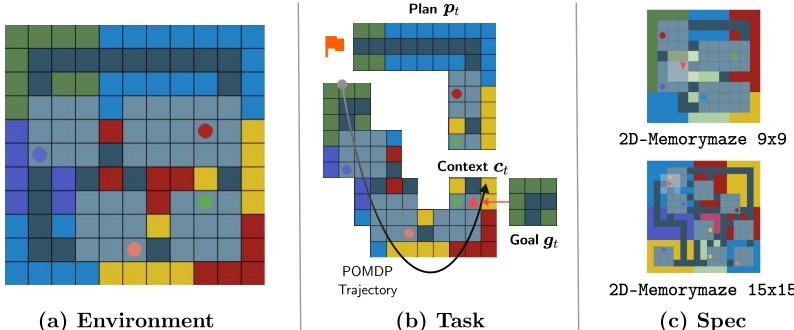

(a) Environment      (b) Task      (c) Spec

Figure 5: **2D Memory Maze**. (a) Procedurally-generated environment. (b) Goal-conditioned navigation task.

on their corresponding floors. To prevent the agent from memorizing the environment, configurations—locations of floors and blocks—are shuffled, resulting in 192 unique setups. We split the training and testing datasets to ensure agents are evaluated on new configurations unseen during training.

Unlike conventional robotics control tasks, the robot cannot perceive the global state of the environment. We restrict the agent's visibility to its own joint information, preventing it from seeing blocks and floors unless the end-effector is close enough to an object to observe and check its color. Otherwise, the agent perceives only its own body. For details of the Blind Color Matching task, see Appendix D.

For the simple superimposed MNIST task, we utilized UNet (Ronneberger et al., 2015) as a backbone, which has been widely used in diffusions (Dhariwal & Nichol, 2021). In advanced tasks, Memory Maze 2D and Blind Color Matching, we utilized Transformer (Vaswani et al., 2017) backbones which showed good performance on diffusion. We conducted an ablation study on the backbone networks in our setting and found that the Transformer performed better despite using fewer parameters.

## 5.2 SUPERIMPOSED MNIST

In the Superimposed MNIST (SMNIST) task, we investigated the impact of varying memory lengths on time complexity and performance while keeping the planning horizon fixed. We designed three SMNIST tasks with memory lengths of 1,568 pixels (1.5k), equivalent to the original 28x28 image resolution. These tasks were augmented to 3,528 (3.5k) and 6,272 (6.2k) pixels to investigate the effects of extended memory in complex pattern recognition tasks. Each task was configured with a fixed planning horizon of 784 steps, equivalent to the original scale. This setup allowed us to assess the computational demands of each baseline model and evaluate how well they retain memory across numerous past observation tokens.

**SMNIST 1.5k** In the 1.5k SMNIST task, the POMDiffuser with Transformer demonstrated the best performance, as it can directly access the context when generating trajectories. The one that utilized Mamba did not generate perfect plans.The one with RNN occasionally failed to reflect the global contextual information. Autoregressive Transformer baselines also failed due to compounding errors; although they could perfectly predict the next token during teacher forcing, they faltered at inference time. Refer to Appendix F.1 for qualitative samples.

**SMNIST 3.5k and 6.2k** For the more challenging 3.5k and 6.2k SMNIST tasks, we compared only the RNN and Transformer memory baselines, as other baselines did not improve performance.

In the RNN-memory baseline, encoding contexts of length 3.5k and 6.2k was computationally too slow. To address this, we reduced the input resolutions by mapping $42 \times 42$ and $56 \times 56$ images to $28 \times 28$ through uniform random sampling of indices. This approach penalized the computational inefficiency of the RNN by limiting its ability to access the full contextual information. For further details, see Appendix B.1.

The Transformer-memory baseline did not suffer from a training bottleneck like the RNN-memory baseline but faced challenges due to memory complexity as the number of contextual tokens increased. As the memory length grew, we had to reduce the batch size quadratically. Consequently, despite consuming the same amount of gradient steps, this model could not converge in a reasonable time.

## 5.3 2D MEMORY MAZE

Table 1: Performance on 2D Memory Maze

| Maze Size | $9 \times 9$ | | | $15 \times 15$ | | |
|---|---|---|---|---|---|---|
| Methods | Maze MSE ($\downarrow$) | Distance ($\downarrow$) | Return ($\uparrow$) | Maze MSE ($\downarrow$) | Distance ($\downarrow$) | Return ($\uparrow$) |
| null-Diffuser | 0.1581 | 4.72 | 0.2159 | 0.1970 | 13.36 | 0.1247 |
| Homogenous Diffuser | **0.0207** | **0.32** | **0.9391** | *Too slow to converge* | | |
| POMDiffuser (RNN) | 0.0506 | 0.492 | 0.8877 | *Too slow to converge* | | |
| POMDiffuser (SSM) | 0.0240 | 0.372 | 0.8973 | 0.0919 | 3.711 | 0.4367 |
| POMDiffuser (TF) | **0.0214** | 0.384 | 0.8545 | **0.0568** | **1.947** | **0.5994** |

In the $9 \times 9$ grid setting, the Homogeneous Diffuser achieved the best performance across all metrics. This model excels by integrating historical context directly into the noisy trajectory during the denoising process, effectively leveraging the benefits of homogeneous modeling in diffusion models. We ensured a fair comparison among models by controlling factors such as batch size (maintaining the same number of gradient steps) and the size of the denoising neural networks. Both SSM-POMDiffuser (SSM) and POMDiffuser (Transformer) performed comparably, demonstrating their ability to handle long sequences effectively.

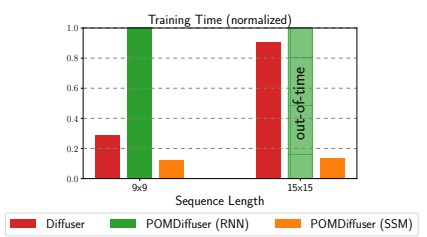

Figure 6: Time comparison in MM2d

In the more challenging $15 \times 15$ grid setting, the Homogeneous Diffuser and POMDiffuser (RNN) models were too slow to converge, making it impractical to obtain results within a reasonable timeframe. The POMDiffuser (SSM) and POMDiffuser (Transformer) models showed decreased performance compared to the $9 \times 9$ grid. This decline can be attributed to the increased complexity and the larger amount of low-level information that must be retained as the grid size expands. Additionally, due to practical considerations, we maintained the same maximum context length, which limited the agent's access to the complete information necessary to reach the goal successfully.

## 5.4 BLIND COLOR MATCHING

In the blind color matching task, which demands extensive memory with a context length reaching up to 3,000, the POMDiffuser (SSM) exhibited superior performance. This task presents significant challenges for the Transformer model, requiring reductions in batch sizes compared to those utilized by the Mamba memory model and resulting in slower convergence rates. On the other hand, Mamba excels in capturing global context, a strength stemming from its ability to recall key high-level information efficiently.

| Method | Return ($\uparrow$) |
|---|---|
| POMDiffuser (SSM) | 0.6956 |
| POMDiffuser (TF) | 0.0187 |

Table 2: Performance on BCM

## 5.5 EFFICIENT PLANNING IN THE LATENT SPACE

**2D Memory Maze $9 \times 9$-LongHorizon** We also observe a slight performance enhancement with the latent level planner as the model size increases while still retaining computational efficiency despite the model's growth. The Homogeneous model, which exhibited strong performance on the 9x9 grid, becomes excessively slow to converge as its time complexity escalates to $O((L + H)^2)$.

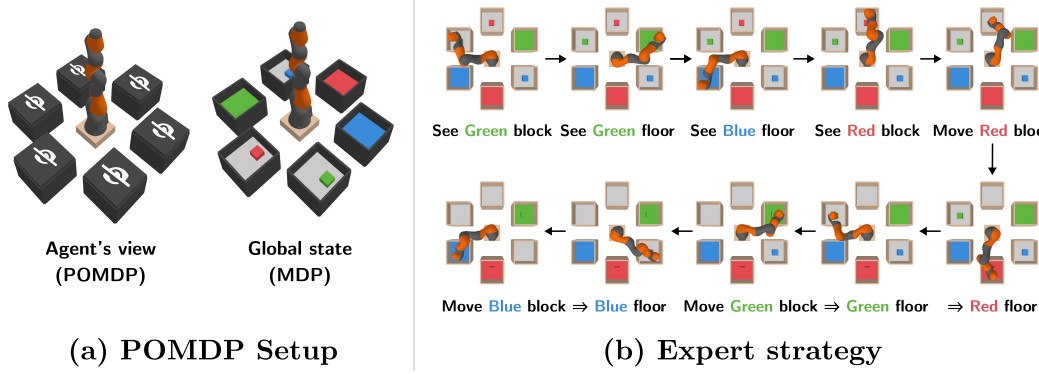

(a) POMDP Setup · (b) Expert strategy

Figure 7: **Blind Color Matching**. (a) Unlike conventional state-based control tasks, the robot agent is not permitted to observe distant parts of the environment from the end-effector. (b) The expert-level strategy for solving the task, which is composed of two phases: exploration and solving the known pair of the block and floor.

Table 3: Ablation studies on latent space planning

| Methods | Maze MSE ($\downarrow$) | Distance ($\downarrow$) | Training Time ($\downarrow$) |
|---|---|---|---|
| POMDiffuser (TF) | 0.0898 | 3.350 | 0.2467 |
| + Latent-Level plan | 0.0766 | 2.937 | 0.1595 |

### 5.6 MEMORIZE BETTER TO PLAN LONGER

Our approach is closely related to predictive coding, as it simultaneously learns contextual memory representations and planning tasks Gregor et al. (2019). We conducted additional experiments on the Superimposed MNIST and Memorymaze-2D datasets to explore the relationship between planning horizon and the amount of global information retained in the belief or memory.

**Superimposed MNIST** Using the Transformer-memory Diffuser-Planner, we compared the attention maps of the Transformer's memory for different plan horizon lengths. We examined the portion of the attention map that contained attention values exceeding a certain threshold across the entire context frame pair, along with qualitative results.

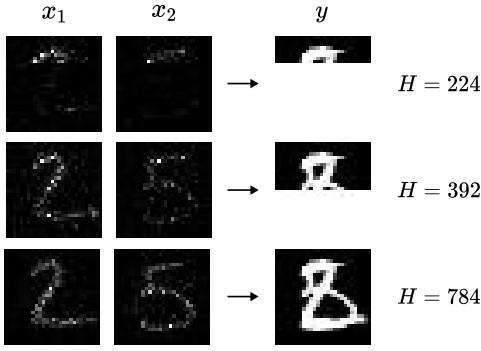

$x_1$   $x_2$   $y$

$H = 224$

$H = 392$

$H = 784$

Figure 8: Attention map across different planning horizons

| Horizon | Global Ignorance ($\downarrow$) |
|---|---|
| 224 | 0.1176 |
| 392 | 0.0284 |
| 784 | 0.0211 |

Table 4: As the planning horizon gets shorter, the belief states tend to be unaware of global information.

**Memorymaze-2D** Using the SSM-memory Diffuser-planner, we carried out a global map probing task, adjusting the planning horizon to 36, 72, 108, and 450 steps. In this task, we utilized a frozen SSM memory model, a component of the SSM-memory Diffuser-planner, to generate a compressed memory representation $\boldsymbol{h} \in \mathbb{R}^D$ from the **context** $\in \mathbb{R}^{N \times C}$. The task involved predicting the top-down global maze layout based solely on this memory embedding. We observed that as the planning

horizon increased, the memory encoded more global information, leading to improved mean squared error (MSE) in predicting the maze layout.

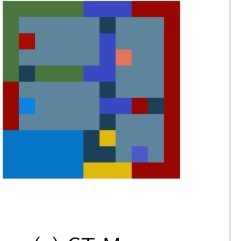

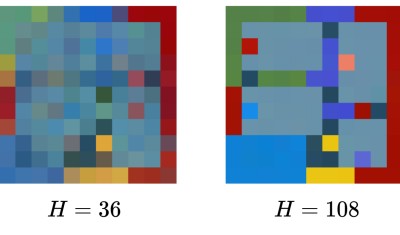

| Horizon | Probing MSE ($\downarrow$) |
|---------|----------------------------|
| 36      | 0.1261                     |
| 72      | 0.0246                     |
| 108     | 0.0174                     |

$H = 36$      $H = 108$

(a) GT Maze      (b) Belief probing results

Figure 9: **Qualitative samples of belief probing.**

Table 5: As the planning horizon lengthens, probing accuracy increases.

## 6   CONCLUSION AND LIMITATIONS

In this paper, we introduced the *POMDiffuser*, the first diffusion-based planning framework designed for POMDPs, addressing the challenge of long-memory and long-planning in partially observable environments. By integrating belief encoding architectures like RNNs, Transformers, and Structured State Space Models (SSMs), our framework extends the capabilities of the Diffuser planner beyond MDP settings. We analyzed the strengths and weaknesses of each approach and demonstrated that POMDiffuser successfully combines long-term memory and extended planning, making it effective for tackling complex tasks in partially observable environments. Additionally, we introduced a new benchmark suite to evaluate diffusion models' long-memory, long-planning capabilities, filling a critical gap in current evaluation methodologies.

Our results show that POMDiffuser offers a powerful and generalizable solution for planning in POMDPs, and we see several future directions to improve upon this work. These include exploring online fine-tuning of the diffusion planner, enhancing belief encoding mechanisms, and further advancing meta-planning capabilities. We hope that our benchmark suite will encourage further research into memory-augmented diffusion models, driving the development of more robust and efficient long-horizon planning solutions in partially observable environments.

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

# A    MODEL ARCHICTECTURES

For the Superimposed Mnist task, we utilized UNet as its backbone, and for 2D Memory Maze and Blind Color Matching, we utilized Transformer.

| Module | Hyperparameter | Tasks | | |
| --- | --- | --- | --- | --- |
| | | Superimposed MNIST | 2D Memory Maze | Blind Color Matching |
| General | Batch Size | 128 | 128 | 24 |
| | Total Steps | 400,000 | 250,000 | 300,000 |
| | Warmup Steps % | 0.05 | 0.05 | 0.05 |
| | Decay Steps % | 0.5 | 0.5 | 0.5 |
| | Max Gradient Size | 0.1 | 0.1 | 0.1 |
| Memory | Input Size | 1 | 1 | 1 |
| | Hidden Dim | 256 | 256 | 256 |
| | Num Blocks | 8 | 4 | 4 |
| | State Size | 16 | 16 | 16 |
| | Expand | 2 | 1.5 | 1.5 |
| Generator | Observation Dim | 1 | 3 (12 if latent) | 22 |
| | Action Dim | 0 | 0 | 0 |
| | Horizon | 784 | 784 | 784 |
| | Transition Dim | 1 | 1 | 1 |
| | Cond Dim | 256 | 256 | 256 |
| | Cross Attention Type | Intermediate | Intermediate | Intermediate |
| | Nhead | - | 8 | 8 |
| | Num Layers | - | 20 | 20 |
| | D Model | 32 | 128 | 128 |
| | Dim Feedforward | - | 512 | 512 |
| | Dim Mults | [1, 2, 4, 8] | - | - |
| | Attention | False | - | - |
| | Num CA Blocks | 3 | - | - |
| | Cond Drop Probability | 0.0 | 0 | 0 |
| | Dropout | - | 0 | 0 |
| | N Timesteps | 1568 | 25 | 25 |
| | Sampling Timesteps | null | null | null |

Table 6: Hyperparameter settings for Superimposed MNIST, 2D Memory Maze, and Blind Color Matching tasks.

## B  ENVIRONMENTS

### B.1  SUPERIMPOSED-MNIST DATA FOR RNN-POMDIFFUSER

When experimenting with SMNIST at resolutions of $42 \times 42$ and $56 \times 56$ to test longer memory sequences, using RNNs becomes computationally impractical due to their sequential processing of long sequence data. To make RNNs more manageable, the data resolution was fixed at $28 \times 28$. However, to maintain a fair comparison with Transformers and SSMs, which can process entire sequences of pixels parallelly, RNNs were restricted to only accessing specific $28 \times 28$ pixel regions from the upscaled $42 \times 42$ and $56 \times 56$ images. This approach ensures a balanced evaluation by compensating for the inherent advantages of models that can handle larger inputs more efficiently.

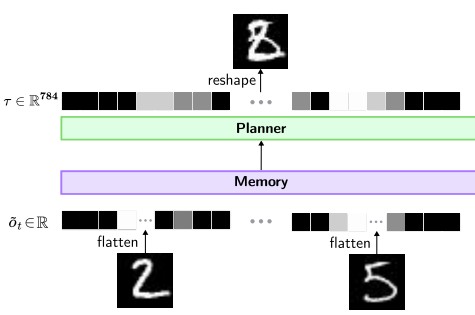

Figure 10: Serialized inputs update belief state for planning.

### B.2  2D MEMORY MAZE

**Environment details.** The agent navigates through a maze from a top-down view but can only see a $3 \times 3$ area around itself, as illustrated in Figure 2. Each element in this grid is mapped to an RGB value. To create a long-horizon, memory-demanding scenario, the agent's observation frame $\mathbf{o}_t \in \mathbb{R}^3$ at each time step is flattened into $\tilde{\mathbf{o}}_t \in \mathbb{R}^{9 \times 3}$. Movement is controlled by four discrete actions: up, down, left, and right.

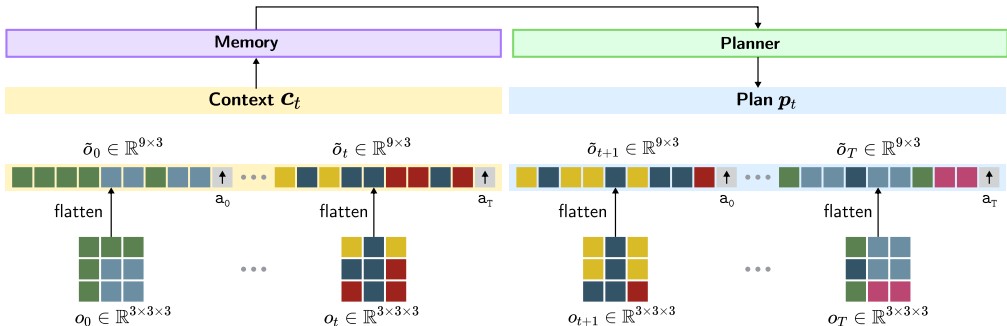

Figure 11: **Preprocessing of observations in Memorymaze-2D dataset**.

**Dataset collection.** We created a scripted policy that navigates the map using a BFS strategy. Every time the agent reaches the goal, the goal location is reset, and the agent continues to explore the map. We randomly select the exploration location from the walkable paths on the map, enforcing that the target navigation location is far from the current position, exceeding a pre-defined L1 distance. We used L1 distance thresholds of 5, 8, and 12 for the Memorymaze-2D 9×9 Long-horizon, Memorymaze-2D 9×9, and Memorymaze-2D 15×15, respectively.

**Training and test split.**  For the training and test split in offline model training and online environment interactions, we generated 5,000 unique maps for training and another 100 maps for testing and validation. For both the Memorymaze-2D 9×9 Long-horizon and Memorymaze-2D 9×9 tasks, we used an episode length of 5,000, while for Memorymaze-2D 15×15, we used an episode length of 10,000.

**Dataset statistics.** To determine the amount of burn-in context required for training our model, POMDiffuser, and for evaluation through environment interactions, we investigated how many contextual frames are necessary to reach any goal point on the map. For Memorymaze-2D 9×9, approximately 100 frames are sufficient to solve any goal location in the maze, while 300 frames are needed for Memorymaze-2D 15×15.

**Evaluation process.** We evaluate the trained POMDiffuser through 100 interactions with test split environments and average the score. More specifically, the agent receives a reward of 1 when it reaches the target location. Since we adopted an open-loop interaction based on the imagined plan, the agent finishes the episode if it incorrectly plans and walks through a wall. The episode then ends, and the agent receives a small reward proportional to how close it was to the target when the episode finished, calculated as:

$$r = \frac{\text{Maze size} \times 2 - \textbf{L1 Distance}(pos_{\text{agent}}, pos_g)}{\text{Maze size} \times 2} \tag{12}$$

## C    BELIEF PROBING IN 2D MEMORYMAZE.

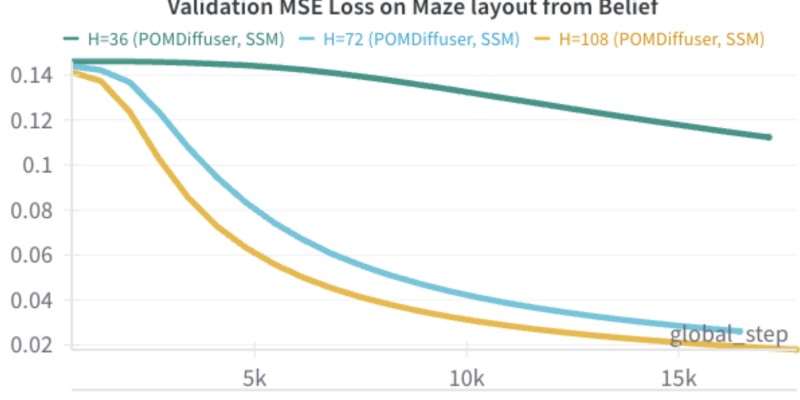

Figure 12: Line plot of validation MSE loss on the maze layout prediction task.

We conducted a probing task to evaluate the informational richness of the belief states used for subsequent planning. After freezing the parameters of the POMDiffuser (SSM), we allowed the model to process some burn-in frames and used the final hidden state to probe the map. We then collected pairs of $(h, maze\ layout)$, and trained a simple Transformer decoder network that predicts the maze layout starting from zero tokens, conditioned on the belief states.

We trained this simple network for 30k gradient steps and compared the MSE loss across different planning horizons of the pre-trained POMDiffuser: 36, 72, and 108.

### C.1    CLASSIFIER-FREE GUIDANCE IN 2D MEMORY MAZE.

We tested Classifier-Free Guidance (CFG), which has shown strengths in conditioned diffusion generative modeling without the need for a separate class classifier. The empirical results did not show a noticeable improvement in the POMDiffuser's generative performance.

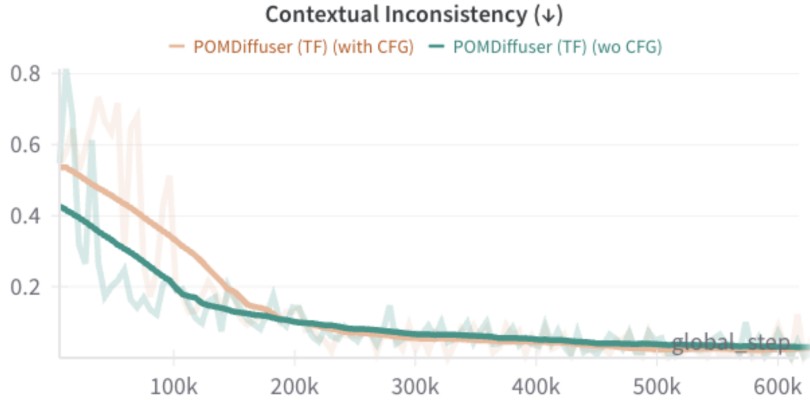

Figure 13: Line plot of contextual inconsistency, measuring how the generated trajectories are misaligned.

## D   BLIND COLOR MATCHING

**Dataset collection.** For dataset collection in Blind Color Matching, we use a PyBullet-based motion planning algorithm to control the Kuka arm with 7 degrees of freedom (DOF) joints. We collected expert policy data, where the robot gathers environment information at the start of each episode and solves the task by picking and placing a block onto a floor tile of the same color as the block. The environment contains 192 unique configurations. We randomized the position of each block and the exploration behavior of the expert robot. Additionally, we incorporated semantic reasoning components into the environment, where blocks and floor tiles of the same color cannot be placed in adjacent spots among the six hexagonal locations. This allows the agent to skip unnecessary exploration by leveraging memory. For example, if 3 out of 6 floor tiles are revealed to be Blue, Green, and Red, and the remaining 3 tiles are unknown, the agent can infer the positions of the remaining blocks without further exploration. If the known tiles are all separated by exactly two spaces, the environment rule that prevents blocks of the same color from being adjacent allows the agent to deduce that the corresponding blocks must be placed on the opposite sides, eliminating the need to explore the remaining tiles.

**Dataset preprocessing.** To convert the MDP state space into the POMDP observation space, we reduced the MDP state size from 43 to 22, using the following format:

- MDP state (size 43):
    - `q_pos (7)`
    - `attachment (3)`
    - `ee_pos (3)`
    - `[cube_pos (3), cube_rotation (4), cube_color (1)] x 3`
    - `floor_color (6)`
- POMDP state (size 22):
    - `q_pos (7)`
    - `attachment (1)`
    - `cube_info (3 + 4 + 3)`
    - `floor_color (4)`

As explained in the limitations, we did not adopt closed-loop re-planning with window slicing. Instead, we used only the last 192 steps for the planning sequence, while all preceding steps were treated as contextual information. The maximum length of the contextual memory is approximately 3,000 steps.

**Training and test split.** We split the total 192 environmental configurations into 180 for training and 12 for testing, ensuring that the agent is evaluated in environments it has not encountered during training.

## E   TRAINING AUTO-ENCODER FOR LATENT-LEVEL PLANNING

We trained a deterministic auto-encoder to demonstrate latent-level planning in the 2D Memory Maze task. While training VAEs is common, we chose to train an auto-encoder to simplify the model design and due to the simplicity of the dataset. We used a shallow 1D convolutional network with residual connections, featuring a bottleneck structure.

Table 7: The Autoencoder architecture used for plan abstraction.

| Layer | Kernel Size | Stride | Channels | Output Channels | Activation |
|---|---|---|---|---|---|
| ResBlock Encoder | 3 | 1 | 3 | 1024 | ReLU |
| Bottleneck Encoder | 3 | 3 | 1024 | 12 | Tanh |
| Conv1D Transpose | 3 | 3 | 12 | 1024 | ReLU |
| Conv1D Projection | 3 | 3 | 3 | 3 | - |

## F   QUALITATIVE RESULTS

### F.1   SUPERIMPOSED MNIST

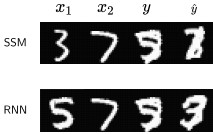

Figure 14: **Randomly Selected Example of the of Superimposed MNIST**.

Our experimental results reveal notable distinctions between the POMDiffuser models employing Structured State Space Models (SSMs) and Recurrent Neural Networks (RNNs) in their ability to maintain and utilize memory for generating consistent outputs. Specifically, the POMDiffuser (SSM) model, when presented with a sequence starting with the digit '3', accurately regenerates the digit '3'. In contrast, the POMDiffuser (RNN) model, under the same conditions, erroneously produces the digit '5'.

Despite these differences in output fidelity, the Learned Perceptual Image Patch Similarity (LPIPS) scores, which quantify perceptual differences between images, do not exhibit significant variation between the two models. This suggests that while both models maintain a perceptual resemblance to the target digit, the SSM-based POMDiffuser demonstrates a superior capacity for updating its belief state with sufficient information to generate the appropriate class. Conversely, the RNN-based POMDiffuser appears less capable of accurately updating its belief state under the same conditions.

This outcome underscores the efficacy of SSMs in capturing and utilizing relevant information to maintain consistency in generative tasks, particularly in environments requiring robust memory and inference capabilities.

## G   ADDITIONAL RELATED WORKS

### G.1   BENCHMARKS FOR LONG PLANNING WITH LONG MEMORY

**Vision-Based Tasks** Vision-based Reinforcement Learning tasks(Mnih, 2013) often feature weak POMDPs, where the problem of partial observability is mitigated by frame stacking or using a simple RNNs network. This approach, however, is not ideal for our experiments, as long-term memory is now always necessary for making optimal decisions. In contrast, benchmarks like DeepMind Lab (Beattie et al., 2016) and Memory Maze (Chen et al., 2021) present challenges that require both reward sparsity and long-term memory (Fortunato et al., 2019). . However, these environments also come with high visual complexity, complicating the direct evaluation of planner-generated trajectories. To evaluate the model effectively, we must either measure the reward from real-environment

interactions or compare the similarity between generated trajectories and real frames by following the generated actions. These CPU-intensive methods, due to the simultor, slow down the discovery of effective models, impeding the speed of the evaluation process.

**Continuous Control Tasks.** Control problems involving continuous state spaces, such as the movement of complex joints, are central to robotics tasks, which is why simulator-based tasks like DMC and Mujoco (**?**Todorov et al., 2012) have been developed. Similarly, benchmarks like D4RL (Fu et al., 2020) have been designed to include various behavioral optimizations for offline RL. However, most control tasks are modeled as MDPs and, therefore, do not require memory. A simple workaround is to transform the MDP into a POMDP by introducing Gaussian noise or delaying perception. However, this approach has limitations, as it still makes encoding the global context in long-term memory optional rather than essential. Robotics tasks, such as AntMaze, Pick and Place, and Block Stacking, are often based on visual observations. However, the increasing visual complexity of these tasks makes the problem more challenging. Furthermore, since they typically use a single map, relying on past observations for memory is optional. As a result, these tasks are not well-suited for evaluating an agent's ability to learn and manage long-term memory.

