# OpenReview forum: "POMDIFFUSER: LONG-MEMORY MEETS LONG- PLANNING FOR POMDPS"
_ICLR.cc/2025/Conference — ICLR 2025 Conference Withdrawn Submission_

### Official Review · Reviewer_UCDJ · 2024-10-31

**Soundness:** 3
**Presentation:** 2
**Contribution:** 1
**Rating:** 3
**Confidence:** 3

**Summary:**

This paper seeks to combine a diffusion approach to planning with a partially observable environment embodied in a POMDP. The new proposed algorithm, POMDiffuser, explicitly encodes memory data into the planner, and is tested on several planning tasks.

**Strengths:**

- planning in partially observable environments is a difficult challenge which generally makes sense to answer using ML/AI methods

- the hyperparameters for each task are acknowledged and their values explicitly written

**Weaknesses:**

In short, I neither understand the problem framework nor details of the solution method. I am not understanding the validation tasks either. I cannot judge the contribution of this paper nor its possible drawbacks. Specific comments are below:

- the paper explicitly positions itself as operating on a POMDP environment, even mentioning it in the abstract. But POMDPs are neither defined nor ever seemingly explicitly used in the paper. The only dynamical model that is introduced is a POMDP only in a very trivial sense: if nothing else, both its transitions and observations are deterministic

- on the topic of the presented dynamical system, the paper calls it a "Structured State Space Model" and says that these are "sequence-to-sequence models well-suited for tasks that require significant memory retention and are particularly effective at processing long sequences due to their computational efficiency". But this model seems to be just a standard Linear Time-Varying (LTV) control system! Control design for LTV systems is challenging, but has certainly been explored since the 1950s or earlier. In general, in the context of agent planning, if these are agent dynamics, I don't see what makes them particularly "well-suited" for any task. After all, the model should not depend on the task: the model is whatever represents the agent's dynamics. If this is simply a learning model, then the agent's dynamics are not ever defined.

- there seems to be a lack of awareness of classical control (let alone planning on POMDPs -- a line of work which is never truly mentioned); the authors call usual linear control systems "time-invariant SSMs" and speak about recent studies that explore the conditioning of system matrices on the input sequence. Again, this is not recent work -- stability of linear systems is a classical introductory control topic

- I do not understand the formal problem that this paper is solving. It does not seem to be ever defined and is mostly just described as "long-term planning". Some questions that come to my mind are: Are there rewards? Is there a reachability task? Does the agent move? What are its dynamics?

- I also do not know what the agent knows about its environment. If its dynamics are just linear *and known*, I don't understand why any learning is necessary: optimal control laws for reward maximization (at least with a particular reward structure) can possibly be derived analytically.

- the details of the proposed solution approach are murky to me. Let me just give one example. Section 3.3 says that "unlike in MDPs, predicting actions solely from adjacent frames in POMDPs can be unreliable". Doing so is not in fact unreliable, it is theoretically impossible: both in general MDPs and POMDPs, there is no unique mapping from a transition (s,s') to an action a that might have caused this transition. To address this issue (and I am not sure what it means to address it, given that the problem simply does not have a solution), the paper says it will use "Transformer encoders". Why? How does that work?

- I do not understand the tasks, which are never truly described (the paper does not even provide a full name for MNIST) -- agent motion, agent knowledge, possible "long-term" rewards, etc. never seem to be defined. The paper says that "our model extends diffusion-based planning models into the realm of meta-learning", but this topic is never discussed.

**Questions:**

While I believe that this paper needs to be *substantially* reworked in order to live to its full potential, a non-exhaustive list of questions that would perhaps clarify some of my understanding are:

- what are the agent dynamics?

- what is the agent task (i.e., what is the formal definition of "long-term planning")?

- what is the agent's knowledge about its environment?

- what are the exact dynamics/tasks/environments in the validation tasks?

---

### Official Review · Reviewer_B6P4 · 2024-11-02

**Soundness:** 2
**Presentation:** 2
**Contribution:** 1
**Rating:** 3
**Confidence:** 4

**Summary:**

The paper presents a diffusion model approach to long-horizon planning in POMDPs, called POMDPDiffusor, by extending existing diffusion models with memory mechanisms, such as RNNs, Transformers, and State Space Models (SSMs). In addition, some benchmarks are proposed, such as Superimposed MNIST (to evaluate the memorization capabilities), 2D Memory Maze (to evaluate navigation in a discrete task), and Blind Color Matching (a robotics task, where blocks need to be placed onto floors with matching colors under partial observability and sparse rewards). The approach is evaluated against itself in these domains.

**Strengths:**

- The paper addresses an interesting and relevant problem in the field of model-based decision-making
- It is mainly well-written, and most parts are easy to follow
- It introduces some new benchmarks that could be interesting for the research community

**Weaknesses:**

**Novelty**

The paper addresses the long-term horizon problem by applying known sequence processing techniques (RNNs, Transformers, SSMs) to known decision-making models, i.e., Diffusors. The paper often refers to the computational complexity of these memory techniques, but these are well-known facts, e.g., RNNs are sequential during training but fast during inference, while Transformers are parallelizable but scale quadratically during inference. Thus, I consider the main contribution as an application of known techniques rather than technical innovation.

The introduced benchmarks seem interesting but their evaluation lacks a comparison with other approaches, which is necessary to assess their suitability for testing long-horizon planning, i.e., Are they sufficiently difficult? Do other approaches really struggle on these new domains, as stated in the paper? See Significance below.

 **Clarity**

The abstract teases generalization as a problem of existing approaches. However, the main challenges addressed in the paper are only focused on long-horizon planning and computational complexity (during training and inference).

 **Significance**

The experimental evaluation of the paper is a pure self-evaluation with POMDPDiffusors without further context.

The paper does a lot of conceptual comparison with prior works, such as world models and alternative diffusion approaches, such as Diffusion Forcing. However, none of these approaches is compared within the experimental evaluation, which leaves many questions open to assess the significance of the work:
1. How does the POMDPDiffusor fare in traditional POMDP benchmarks like Pocman, Battleship, etc., compared with prior approaches?
2. Do prior approaches really scale that badly, as stated in the paper? We need to see the numbers - not only the words
3. Do prior approaches really struggle in the new benchmark domains, i.e., are the benchmark domains really justified? Again, we need the numbers - not only the words.

Without any further evidence regarding these questions, the true advancement of the work remains unclear.

**Minor**

- At the end of page for a UNet model is refered to out of nowhere which has not been mentioned and explained before.
- In Related work a reference is missing in "Efficient World Models"

**Questions:**

1. In the 2D Memory Maze experiments, what is the implementation difference between Diffuser and POMDPDiffuser? As stated earlier in the paper, Diffuser was only designed for MDP settings.

---

### Official Review · Reviewer_q1Wk · 2024-11-06

**Soundness:** 2
**Presentation:** 2
**Contribution:** 2
**Rating:** 3
**Confidence:** 3

**Summary:**

This paper proposes a method in terms of achieving both long-memory and long-planning capabilities from past histories in POMDP, which uses diffuser models for memory utilization and long-term planning in complex environments. This method adopted Diffuser-based models which addressed the autoregressive planning problems existing in previous models like RNN, Transformers, and SSMs, it also improved over former diffuser models by extending its use to POMDPs. The authors also proposed a new benchmark suite to evaluate long-memory and long-planning capabilities within the Diffusion framework.

**Strengths:**

1. This paper proposes a method to extend the diffuser planner to POMDPs.

**Weaknesses:**

1. This paper claims to improve upon SSMs, RNNs, and Diffusers; however, it primarily integrates these models by using an SSM as the memory encoder and a diffusion model for planning with the memory.
2. It does not address the issues associated with transformers as stated in the introduction. The model still relies on a transformer encoder for action selection, predicting the full action sequence from past trajectories.
3. The new evaluation benchmark does not appear to provide enough innovation to be considered a genuinely new benchmark.

**Questions:**

1. Could the author provide clearer explanations on how their framework differs from SSMs, RNNs, Transformers, and Diffusers beyond simply incorporating them into different parts of the framework?
2. Could the author compare their proposed benchmark to existing MNIST and Maze environments to better illustrate how it differs?

---

### Official Review · Reviewer_ksWH · 2024-11-06

**Soundness:** 2
**Presentation:** 1
**Contribution:** 1
**Rating:** 3
**Confidence:** 2

**Summary:**

This work considers offline RL in partially observable environments using state-space models as history representations and diffusion models as policy model.
The result is trained in supervised manner (behavior cloning).

It introduces three (new) tasks - based on MNIST, a grid problem, and a pick-and-place task - and provide an ablation study on their model.
The ablation is against transformer and RNN-based history representations, as opposed to state-space model.

To my best understanding, there are no theoretical contributions claimed made in this paper.

**Strengths:**

The problem of offline RL is difficult and important, and should be of relevance to a significant part of the ICLR community.
Additionally, given the success of diffusion models (including in policy generation), it make sense to further investigate their capabilities, limitation, and applicability.
Especially progress in tackling partially observable environments is important, as they are ubiquitous in the real-world yet avoided due to their complexity, and novel generative (sequential) approaches look like a reasonable approach.

**Weaknesses:**

The paper is difficult to understand and the contributions are not quite fleshed out.

As someone who is not particularly familiar with the background (in particular, the "Diffuser", I suppose?), it is difficult for me to infer exactly what the contribution is and how it works.
In particular, the text is currently imprecise both in English as well as math.
Examples include:
- The proposed method to "model memory" is explained as "through cross-attention computation during the denoising process", and otherwise does not seem to give any details.
- It is claimed that, by "separating memory and planning", the complexity reduces from one O notation to the other, but unclear where these come from.
- The state-space is defined as transforming an input x in R^{T x D} to output y in R^{T x D} (where x and y have the same size) but it is not quite ever really clear what x and y would be for the POMDPDiffuser (most likely due to lack of my background).

This makes me believe (perhaps wrongly) that the proposed method is a combination of supervised learning of state-space models to represent histories and diffusion models to learn policies for these histories from decision data.
This, without additional contributions - which may be there but not understood - seem to reduce to behavior cloning on sequences, which is somewhat lacking in novelty.

Lastly, the experimental evaluation does not seem to be very convincing.
In particular, there seem to be no baselines, other than ablations on their own approach, and I must assume offline RL methods for POMDPs exist (it was not claimed otherwise in the paper).

**Questions:**

N/A

---

### Official Review · Reviewer_6XiC · 2024-11-07

**Soundness:** 3
**Presentation:** 3
**Contribution:** 2
**Rating:** 6
**Confidence:** 4

**Summary:**

The paper introduces POMDiffuser, a diffusion-based planning framework designed for POMDPs. The aim was to extend diffusion models to handle long-term memory and long-horizon planning in POMDP settings. They incorporate various (belief) encoding architectures, including RNNs, Transformers, and Structured State Space Models, and evaluate their performance on newly proposed benchmarks.

**Strengths:**

- Addressing long-term memory and planning in POMDPs is a significant challenge in reinforcement learning and decision-making.
- The proposal of a new benchmark suite for evaluating diffusion models in POMDPs was very interesting and could be valuable to the research community.
- Investigating different memory architectures (RNNs, Transformers, SSMs) provides insights into their trade-offs in the context of diffusion planning.

**Weaknesses:**

- The paper lacks a solid theoretical analysis explaining why diffusion models are suitable for long-memory and long-planning tasks in POMDPs. There is no discussion on the convergence properties, limitations, or potential pitfalls of applying diffusion models in this context. As a reader, I was hoping to see it atleast in the appendix section of the paper.
- The paper acknowledges that the proposed method struggles with more complex tasks but does not delve into why this is the case. I would suggest adding a section/few lines on how it might be addressed in future work.
- The experiments seem to be very simplistic.

(Follow up weaknesses in the Questions section)

**Questions:**

- Can the author(s) provide theoretical analysis of how memory length affects planning horizon in your framework?
- What are the convergence guarantees for POMDiffuser, especially when dealing with very long sequences? -- this is something that I am interested in learning more about.
- (minor) How does the belief state representation quality degrade over longer horizons?
- There seems to be a very big performance gap in Blind Color Matching (0.6956 vs 0.0187) between SSM and Transformer variants is striking. Could you provide an analysis of why this occurs? How does this change with different Transformer architectures/configurations?
- (minor) Have you explored any techniques to reduce computational complexity while maintaining performance?
- (minor) This is an interesting framework, how might this be extended to multi-agent POMDP settings?

---

### Official Review · Reviewer_uRby · 2024-11-10

**Soundness:** 3
**Presentation:** 2
**Contribution:** 3
**Rating:** 3
**Confidence:** 3

**Summary:**

The authors extend diffusion-based planning to POMDPs with sparse rewards using memory. A heterogeneous approach based on cross-attention is adopted to incorporate memory, enabling an $O(L \log L + H^2)$ complexity instead of $O(L^2 + H^2)$ where $L$ is the memory length and $H$ is the planning horizon.  More efficiency is achieved using inverse dynamics and latent-level planning for long horizons.  Three POMDPs are proposed to test the proposal.

**Strengths:**

- Proposes a heterogeneous approach to modeling memory in diffusion-based planners for POMDPs.
- Proposes three POMDPs to evaluate the proposed approach: Superimposed-MNIST, 2D Memory Maze (MM2d), and Blind Color Matching (BCM)
- Compares different configurations of the proposed design against baselines along with some ablations.

**Weaknesses:**

- The writing is rushed and a bit disconnected.
- Contributions mainly take the form of the empirical results presented, comparing different configurations of known techniques, without new theoretical/algorithmic insights.
- Given the status of the writing, it's difficult to appreciate the empirical results without significant effort - I'm reading the experiments section without fully understanding the methodology and I have to keep going back to the (rushed) prior sections.
- I seems unlikely those serious issues with the presentation can be addressed without a major revision.

**Questions:**

Presentation
==========
- Abstract
  - Needs a few iterations to improve focus.
  - Didn't seem relevant to mention how humans use long-term memory or meta-learning.  It seems there was no further elaboration on those themes later in the paper.
  - Both "Diffusers" and "diffusion-based planning models" are used.  Prefer the latter.
  - It's not clear what "conventional Diffuser models" refer to, and the claim that they "often memorize specific environments" was not justified in the main text (correct me if I'm wrong - L255 was relevant, but doesn't discuss this specific claim).  Is this claim necessary for the abstract?
  - Last two sentences seem to trail off rather than stating the main contributions clearly.
- S1 - Introduction
  - First sentence is a bit problematic as a very broad statement.  Please consider revising.
    - The notions of "effectively" and "memorize" were not defined.
    - Last sentence in 1st paragraph says "leveraging past experiences" which is more general than "memorize", so prefer the former.
  - L74: The wording here is confusing "performs well in tasks requiring **complex** reasoning .. struggled with more **complex** planning tasks".  Please rewrite for clarity.
- S3 - Memorize to plan
  - Recommend to lead with an introductory sentence.  The first line in S3.1 seems suitable.
  - L153: writing gets a bit rough.  Please rewrite.
  - Please surface sparse rewards in the introduction as the main focus;  it was only mentioned in passing on L036 vs L161.
  - L185: please explain how truncating the trajectory is performed given the sparse reward situation.
  - L189: please introduce homogeneous vs heterogeneous memory architectures.  The current writing assumes the reader is already familiar with those notions.  It would help to also cite examples of each approach.
  - L208: where is $\beta$?
  - L209-210: This seems more like a footnote since Superimposed-MNIST is yet to be introduced.
  - L222: please qualify and justify the claim that using adjacent frames only in POMDPs is unreliable.
- S5 - Experiments
  - L352: Is there an appendix with this ablation study?
  - Some tables and/or figures were not referenced in the main text.  Please fix.

Nitpicking
========
- L159-160 + L253 and elsewhere: please use the correct citation style.
- L220: What is Tedrake?  Is this a misformatted citation?
- L242: Missing citation

---

### Note · Authors · 2024-11-26

I have read and agree with the venue's withdrawal policy on behalf of myself and my co-authors.